# Enhancement of Antioxidant Property of N-Carboxymethyl Chitosan and Its Application in Strawberry Preservation

**DOI:** 10.3390/molecules27238496

**Published:** 2022-12-02

**Authors:** Luyao Bian, Huigang Sun, Ying Zhou, Yang Tao, Chong Zhang

**Affiliations:** 1College of Food Science and Technology, Nanjing Agricultural University, Nanjing 210095, China; 2College of Food and Biological Engineering, Xuzhou University of Technology, Xuzhou 221018, China

**Keywords:** bacterial laccase, N-carboxymethyl chitosan, gallic acid, grafting, strawberry preservation

## Abstract

Bio-enzymatic grafting phenolic acid to chitosan derivative is an efficient and environmentally friendly molecular synthesis technology. In the present study, N-carboxymethyl chitosan (CMCS) was grafted with gallic acid (GA) using recombinant bacterial laccase from *Streptomyces coelicolor* as a catalyst. GA and CMCS were successfully grafted as determined by measuring amino acid content, Fourier transform infrared (FTIR) spectroscopy and ultraviolet-visible (UV-Vis) spectroscopy. Then, the effect of GA-g-CMCS coating on the freshness of strawberries at 20 ± 2 °C was explored. The physiological and biochemical quality indicators of strawberries during storage were monitored. The 1.5% GA-g-CMCS coating helped to protect the antioxidant properties and nutrients of strawberries and extend the shelf life. Specifically, it reduced the weight loss of strawberries during preservation (originally 12.7%) to 8.4%, maintained titratable acidity content (TA) residuals above 60% and reduced decay rate from 36.7% to 8.9%. As a bioactive compound, GA-g-CMCS has the potential to become an emerging food packing method. These results provide a theoretical basis and reference method for the subsequent synthesis and application of CMCS derivatives.

## 1. Introduction

With globalization and growing of consumer demand, fruit is circulating on a large scale, taking longer to transport and store. Rapid changes in the structure and biochemical properties of fruits after harvest may accelerate fruit decay and nutrient loss [1]. Nutrient loss during storage of fruits is mainly caused by oxidase and reactive oxygen species (ROS) [2]. ROS is a reactive substance that can cause damage to biomolecules, including proteins, lipids and nucleic acids [3]. Fruits have antioxidant defenses to protect important biomolecules from damage from ROS, such as superoxide dismutase (SOD) [4]. However, ROS buildup exacerbates cell damage due to the continuous depletion of antioxidants during storage [5]. This makes it challenging to maintain fruit quality during transportation and storage.

Strawberries have high nutritional value that may help reduce the risk of cardiovascular events due to their rich ellagic acid and flavonoids, as well as powerful antioxidant properties [6]. But strawberries are not easy to store due to low firmness. If stored improperly, the loss of ascorbic acid and anthocyanin in strawberries will be aggravated, and the antioxidant activity in vitro will be negatively affected [7]. Therefore, it is urgent to develop packaging materials that can delay the loss of antioxidant substances in strawberries.

The chemical residues and low biodegradability of traditional petroleum-based packaging materials used in food have caused serious damage to the environment and humans [8,9]. Active edible coatings are a promising alternative to extending the shelf life of fruits [10]. Coating treatment can maintain harvest freshness by reducing moisture loss, respiration rate, gas exchange and oxidation reaction rate [11].

Carboxymethyl chitosan (CMCS) is a water-soluble derivative of chitosan (CS) that is used in biomedicine and environmental remediation because of its good degradability and biocompatibility [12,13,14,15]. Compared with the weak ultraviolet blocking performance and high-water vapor transmission rate of CS film [16], CMCS with good plasticizing effect on the film is a strawberry storage packaging material with more potential [17]. CMCS mainly includes O-CMCS and N-CMCS [18], and the hygroscopicity of N-CMCS is lower than that of the former, which is more conducive to fruit preservation [19].

As a material that is easy to modify, the antioxidant capacity of CMCS can be enhanced by grafting functional substances such as phenolic acids [20,21]. Currently, there are four main methods for phenolic acid grafting, including carbodiimide coupling, enzymatic grafting, free radical-mediated grafting and electrochemical methods [22]. Compared with other methods, enzymatic grafting is simple, safe, environmentally friendly [23] and relatively low cost [24]. 

Laccase (EC 1.10.3.2) is a copper-based polyphenol oxidase that efficiently catalyzes the oxidation of phenols, aromatic amines and aliphatic amines [25,26]. It is an environmentally friendly biocatalyst that only requires oxygen as a co-substrate and releases water as the only by-product [27]. Laccase plays key roles in various areas of food industries, including beverage processing, ascorbic acid determination, baking, as biosensor and to improve food sensory parameters [28,29]. Thus, the preparation of fruit preservation materials catalyzed by laccase is a promising, effective and safe method.

At present, there is no research on the use of laccase to catalyze the grafting of phenolic acid and CMCS in food preservation. Therefore, we wanted to use gallic acid (GA) as the model phenolic acid to explore the feasibility of CMCS complex applied to fruit preservation.

In this study, GA was grafted onto N-CMCS (GA-g-CMCS) with small laccase (SLAC) derived from *Streptomyces coelicolor*. In order to explore the effect of GA-g-CMCS coating on strawberry preservation, the physiological and biochemical indicators of strawberry were measured. This research aims to develop new, safe and bio-based packaging materials prepared from CMCS and phenolic acids for emerging food processing. The results will provide theoretical basis and reference for the further synthesis and application of bioactive compounds.

## 2. Results and Discussion

The commonly used method of conjugation synthesis of phenolic acids, CS and CMCS is to add chemicals to initiate polymerization of hydroxyl or carboxyl radicals in the system [22]. In the field of food processing, safety and environmental protection are a key concern for consumers. Therefore, laccase-catalyzed phenolic acids and CMCS conjugated products for food packaging are more acceptable to consumers.

### 2.1. Characterization of GA-g-CMCS

#### 2.1.1. Amino Content

CMCS and GA were grafted with SLAC to obtain a stable brown product, which became a yellowish powder after lyophilization. The absorption value of GA-g-CMCS at 570 nm was 0.89, and that of CMCS was 1.23. Lower absorption value means lower residual amino group. As shown in Figure 1A,B the decrease in amino content may be due to the formation of covalent bonds by GA grafted to the free amino group of the CMCS via laccase [30].

#### 2.1.2. UV-Vis and FTIR Analyses

In order to further explore the grafting of GA and CMCS, the spectroscopic properties were determined. The ultraviolet-visible (UV-vis) spectra of CMCS and GA-g-CMCS are shown in Figure 2A. There was no absorption peak in the UV-vis spectrum of CMCS, which may be due to the lack of chromosphere. However, GA-g-CMCS displayed two distinct peaks at 213 nm and 260 nm. This result is consistent with that of GA grafted with O-CMCS [31]. This difference could be attributed to the introduction of the benzene ring of GA [32], confirming the successful reaction between GA and CMCS. 

The Fourier transform infrared (FTIR) result showed that GA-g-CMCS had many new peaks compared to CMCS (Figure 2B). There were five smaller absorption peaks at 3444, 3369, 3125, 3068 and 2885 nm, which were caused by the stretching vibration of phenolic ring -OH in GA [33]. The absorption peak at 1258 nm was generated by the stretching vibration of -COOH in GA. There was a strong absorption peak at 1053 nm caused by the twisted vibration of the -C-N- bond in CMCS. There were five small shoulder peaks between 1000–750 nm, generated by the shaking vibration of H_2_O. This indicated that although the GA-g-CMCS had been freeze-dried, they were not completely dried and retained a small amount of water. At the same time, it was confirmed that CMCS and GA did indeed crosslink to form a new polymer. 

#### 2.1.3. Antioxidant Activity

As shown in Figure 3, the 2,2-diphenyl-1-picrylhydrazyl (DPPH) radical scavenging capacity of samples increased with increasing concentration. When the concentration was increased from 0.5 mg/mL to 2 mg/mL, the DPPH radical scavenging capacity of CMCS and GA-g-CMCS increased to 4.5% and 5.4%, respectively. At the same concentration, the value of GA-g-CMCS was higher than that of CMCS. This indicated that GA-grafted CMCS has better DPPH radical scavenging ability than CMCS. The results are consistent with the reports [31,34] that the DPPH radical scavenging ability is positively correlated with GA contents in CMCS.

### 2.2. Coating Strawberries with GA-g-CMCS

As shown in Figure 1B, the strawberries were coated. By measuring the physiological and biochemical indicators changes in a constant temperature and humidity room (20 °C, RH 50%) for 4 days, the effect of coating treatment on strawberry freshness was explored.

### 2.3. Physiological Analysis of Strawberries

#### 2.3.1. Respiration Rate

The change in respiration rate of strawberries during storage is shown in Figure 4A. A change in color from orange to purple indicated that the respiration rate has changed from weak to strong. With the prolongation of storage time, the respiration rate first increased and then decreased, reaching the maximum on the second day of storage. Because the strawberries were not fully ripe, a post-ripening effect occurred during storage, resulting in an increased respiration rate. When the strawberries were fully ripe during storage, the respiration rate decreased. 

Figure 4A also showed that the respiration rate of different treatments strawberries differed for the same storage time. The respiration rate of the control group was much higher than in the other treatment groups, indicating that the control group consumed too many nutrients, while there was no significant difference between the coated groups. Therefore, the coating treatment can effectively inhibit the respiration rate of strawberries and reduce their loss of nutrients.

#### 2.3.2. Firmness

Fruit firmness is one of the important indicators for evaluating fruit ripeness and storage quality [35]. During the ripening and aging of the fruit, the firmness gradually decreases. Firmness reflects the degree of ripening and softening after ripening, and can therefore provide guidance for the proper storage of the fruit. Figure 4A,B show that the firmness of strawberries in all treatment groups decreased with longer storage time. In the same storage time, the control group had the lowest firmness and the most severe degree of softening, while the 1.5% treatment group had the highest firmness. 

A decrease in the firmness of strawberries may occur after ripening or by respiration. It was found that respiration rate and loss of firmness of the coated strawberries were both relatively slowed down compared to the uncoated. The film formed on the surface after coating may inhibit normal respiration, resulting in a slowdown in firmness and delay the ripening of strawberries. Therefore, treating strawberries with GA-g-CMCS coating solution of 1.5% has a good freshness preservation effect and is conducive to storage.

#### 2.3.3. Weight Loss Percentage

Figure 4B shows the change in the weight loss percentage (WLP) of strawberries during storage. A change in color from green to red indicated that WLP changed from low to high. WLP for all groups of strawberries increased gradually throughout the storage period. However, the WLP values in the treatment groups were lower than that in the control group. At the end of storage, the control group had the highest WLP (12.7%), while strawberries treated with 1.5% GA-g-CMCS had the lowest WLP (8.4%). This is superior to the WLP of strawberries treated with genipin-crosslinked N-2-hydroxypropyl-3-butyl ether-O-carboxymethyl chitosan (HBCC) film (about 13%) [36], and is basically the same as the WLP of strawberries with preservatives that hybridize CMCS with metal–organic frameworks (MOFs) [37], but the cost of GA-g-CMCS is relatively lower.

WLP in strawberries is mainly associated with respiration and moisture evaporation through the peel of the fruit. The rate at water loss depends on the water pressure gradient between the fruit tissue and the surrounding atmosphere and the storage temperature [38]. The coating is a selective barrier that alters the internal atmosphere and helps to retard the respiration rate of fruit, thereby reducing WLP [39].

#### 2.3.4. Decay Rate

Figure 4B,C shows that the decay rate of strawberries increased with the extension of storage time, and the control group was significantly higher than that of the treatment groups. At concentrations of 1.0%, 1.5% and 2.0%, the decay rates of GA-g-CMCS coated strawberries were 10.0%, 8.9% and 12.2%, respectively, which were higher than that of 2% CMCS. CMCS induces the activity of defense-related enzymes, causing plants to produce disease-resistant substances such as phenols and participate in defense mechanisms, thereby delaying decay [40]. After combining GA, its resistance to decay was strengthened.

#### 2.3.5. Color

*L* represents the whiteness of the sample, with a larger value indicating whiter; *a* represents the redness/greenness of the sample, with larger value indicating redder; and *b* represents the yellowness/blueness of the sample, with larger value indicating yellower. The chroma of strawberries from different treatment groups during storage was characterized, and the values of *L*, *a* and *b* values are given in Figure 5. Obviously, the value of *L* does not change significantly, and the values of *a* and *b* decreased with storage time for coated strawberries, while they showed much weaker change for the control group. 

After storage, the ΔE of the control group was about one-third that of other treatment groups. This indicated that the coating treatment had little effect on the maintenance of the color of the strawberries. However, the color of the strawberry skin in the control group in Figure 5E was darker. Therefore, the measured data characterized the overall color change of the strawberry. This showed that the effect of coating treatment on the surface color of strawberries was small. The reason for the darkening color inside the strawberry pulp remains to be explored.

### 2.4. Biochemical Analysis of Strawberries

#### 2.4.1. Soluble Solids Content

Soluble solids content (SSC) is one of main parameters for evaluating the quality and nutritive value of strawberry, which is related to consumers’ taste preferences [41,42]. In general, the SSC of fruit increases gradually during maturation, but may decrease during aging. Therefore, SSC is an important indicator of good storage resistance. 

With the increase of storage time, the SSC of strawberries in all groups decreased (Figure 6A). SSC residues were slightly higher in the treatment groups than in the control group. Strawberries treated with 2.0% GA-g-CMCS had the highest SSC content at the end of storage, which means that treatment with GA-g-MCS coating can effectively delay aging. As suggested by Ali et al., slow breathing delays metabolite synthesis and use, resulting in lower SSC [43]. This is also consistent with the results of Figure 4. 

#### 2.4.2. Titratable Acidity Content

The content of organic acids in strawberries has an important impact on their taste, flavor, sugar acid ratio, pH and processing properties [44]. The change in titratable acidity content (TA) content of strawberries during storage is shown in Figure 6B. The TA content decreased continuously during storage due to the consumption of organic acids by the physiological activities of strawberries. The TA content of the treatment groups was higher than that of the control group, indicating that the coating treatment could inhibit the respiration of strawberries, thereby reducing nutrient losses. 

The 2.0% CMCS and 1.5% GA-g-CMCS coating treatments had the best protective effect on the TA content of strawberries. After coating, the residual amount of TA exceeded 60%. Strawberries treated with chitosan-whey protein isolate coating lost about 58% TA after similar storage conditions [45], indicating that GA-g-CMCS is better for protecting strawberry TA.

#### 2.4.3. Ascorbic Acid Content

Ascorbic acid (AsA) can be used as a key indicator of oxidative degradation of fruits and is also important for human health [46]. AsA has been reported to have the ability to scavenge superoxide and hydroxyl radicals, as well as to regenerate α-tocopherol [47]. 

Figure 6C shows that the AsA content of all strawberries gradually decreased during storage. Under the same storage time, the AsA content of the control group was lower than that of the other treatment groups. The coating treatment of 1.5% G-G-CMCS significantly reduced the loss of AsA in strawberries. After 4 days of storage, the remaining amount of AsA was 53.8%, which was similar to the results of strawberries treated with carboxymethyl cellulose with chitosan composite coating [48]. However, the respiration intensity of the latter was not effectively inhibited, and the loss of AsA may increase with the extension of storage time. Combined with Figure 4, it can also be found that AsA was significantly reduced on the second day when strawberry respiration rate was highest.

Fruit preservation is a process of compound regulation. The coating acts as a protective layer and controls the permeability of O_2_ and CO_2_, thereby reducing the autoxidation potential of the fruit, which may avoid further exacerbation of the protein damage [49]. GA-g-CMCS can effectively retain antioxidants, which is conducive to maintaining a complete cell structure and reducing the rot rate of fruits. 

#### 2.4.4. Reduced Glutathione Content

The change in the GSH content in strawberries during storage is shown in Figure 6D. This suggests that the content of GSH decreased overall but fluctuated as storage time increased. This dynamic change may be due to the fact that glutathione is involved in the detoxification of reactive oxygen species in fruits [50]. As storage time prolongs and reactive oxygen species accumulate, strawberries continue to consume and produce glutathione to maintain cell viability. This may be caused by the constant consumption and production of GSH in the strawberries during storage.

#### 2.4.5. Catalase, Ascorbate Peroxidase and Superoxide Dismutase Activity

Changes in antioxidant enzymes (CAT, APX and SOD) in strawberries are shown in Figure 7. CAT activity in strawberries decreased with storage time, and all treatment groups had smaller decreases than those in the control group (Figure 7A). The CAT of strawberries treated with 1.5% G-G-CMCS coating was superior to other groups. This may be due to the deterioration of the quality of strawberries during storage, resulting in the production of hydrogen peroxide. CAT reduces the damage of hydrogen peroxide to cells, thereby reducing its enzyme activity. 

Antioxidant enzymes play a very important role in inhibiting oxidative stress [51]. Strawberries constantly produce substances that are harmful to cells, such as hydrogen peroxide. To reduce cell damage, strawberries need to produce more antioxidant enzymes to remove harmful substances. GA-g-CMCS had higher activity of CAT, APX and SOD at the end of storage than the control and CMCS groups, indicating that this coating treatment was effective for the preservation of strawberries.

## 3. Materials and Methods

### 3.1. Materials

Recombinant *Escherichia coli* strain BL21 (DE3) harboring plasmids pET-23a (Novagen, Darmstadt, Germany), containing the SLAC gene from *Streptomyces coelicolor* A3 (2) (GenBank No. NC_003888.3), were constructed previously [52]. The SLAC gene was expressed through Isopropyl-β-D-thiogalato-pyranoside (IPTG) (100 μM) induction, and the laccase was purified with Ni-NTA Super flow Cartridges (Sangon Biotech, Shanghai, China) as previously reported [53]. The activity of SLAC was measured at 45 °C for using 2,2-azino-bis (3-ethylbenzothiazoline-6-sulfonic acid) (ABTS) as the substrate. One unit of activity was defined as the amount of laccase required to oxidize 1 µM ABTS per min.

IPTG, ABTS and DPPH were purchased from Sigma Aldrich (St. Louis, MO, USA). CAT, APX and SOD activity detection kits were purchased from Beijing Solarbio Science & Technology Co., Ltd. (Beijing, China). The high-viscosity N-carboxymethyl chitosan (C_8_H_14_NO_6_) (degree of deacetylation ≥ 85%, 220 Mw) and GA (purity ≥ 98%) were purchased from Shanghai Macklin Biochemical Co., Ltd. (Shanghai, China). The strawberries (variety: Beauty) were purchased from the Nanjing fruit farm (Nanjing, China). All other chemicals were standard reagent grade.

### 3.2. Preparation of GA-g-CMCS

The enzymatic synthesis of GA-g-CMCS was performed based on the heterogeneous grafting method with some minor modification [38]. CMCS powder and GA were dissolved in phosphate buffer (pH 6.5) and methanol, respectively. Then 40 mL CMCS solution (25 mg/mL) was mixed with 5 mL of GA solution (4 mg/mL), and 5 mL of SLAC (5 U/mL) was added. The mixture was reacted with continuous stirring (100 rpm) at 40 °C for 1 h and then placed in a boiling water bath for 10 min to inactivate SLAC. The product was collected by centrifugation for 15 min (8000× *g*, 4 °C) with a centrifuge (Thermo Fisher Scientific, Waltham, USA). To remove any free GA, the product was washed with ethanol and water separately and centrifuged. GA-g-CMCS were prepared by lyophilizing the washed product and stored at 4 °C for later analysis.

### 3.3. Characterization of GA-g-CMCS

#### 3.3.1. Determination of Amino Content

The amino contents of CMCS and GA-g-CMCS were determined based on a reported method with slight modification [54]. CMCS and GA-g-CMCS were separately dissolved in deionized water to a concentration of 1 mg/mL. A total of 2 mL of ninhydrin solution (50 mg/mL, ninhydrin in dimethylformamide) and 0.5 mL of acetate buffer (0.2 M, pH 5.5) were added to 0.5 mL of the sample solution. After reacting in boiling water for 30 min, the samples were cooled to room temperature. The absorbance at 574 nm was measured to compare changes in amino content of CMCS and GA-g-CMCS using a UV-2450/2550 spectrophotometer (Shimadzu, Kyoto, Japan).

#### 3.3.2. UV-Vis and FTIR Analyses

The CMCS and GA-g-CMCS powders were individually dissolved in deionized water at a concentration of 0.5 mg/mL. The UV-vis spectrum was recorded by scanning samples from 200 to 600 nm with a spectrophotometer (Shimadzu, Kyoto, Japan).

The Fourier transform infrared (FTIR) spectrum of samples was determined by Nicolet iS50 spectroscopy (Madison, WI, USA) in the frequency range of 4000–500 cm^−1^. Each sample (1 mg) was mixed with KBr (100 mg) and ground evenly by agate mortar. The grafting situation of the GA and CMCS was determined by bond analysis.

#### 3.3.3. Antioxidant Activity

The antioxidant activity was assayed using the scavenging activity of the DPPH radical based on a reported method [55]. A volume of 2 mL of DPPH ethanol solution (0.1 mM) was mixed with 2 mL of sample solution with different concentrations (0.5–2.0 mg/mL). After full shaking, the sample was incubated for 30 min at room temperature in the dark. The absorbance of the sample was measured at 517 nm with a spectrophotometer (Shimadzu, Kyoto, Japan). The control used deionized water instead of a sample solution. The DPPH free radical scavenging capacity was calculated as follows:(1)$$\mathrm{DPPH scavenging ability}\left(\%\right)=\left(1-A_{\mathrm{sample}}/A_{\mathrm{control}}\right)\times 100\%$$
where A_sample_ is the absorption value of the sample at 517 nm and A_control_ is the absorption value of the control. 

### 3.4. Coating Strawberries with the CMCS and GA-g-CMCS 

The strawberries were washed in physiological saline and then selected for uniformity of size, shape and color. Any fruit with defects, injuries or blemishes was discarded. The strawberries were randomly divided into 5 groups, and the control group was uncoated. The treatment groups were immersed in solutions of CMCS (2.0%, *w/v*), and GA-g-CMCS (1.0%, 1.5% and 2.0%, *w/v*) at 25 °C for 1 min. After drying with cold air, the strawberries were stored in a constant temperature and humidity chamber (Stik, FL, USA) at 20 °C and RH 50% for 4 days. Samples were taken daily until the end of storage. All experiments were performed in triplicate.

### 3.5. Physiological Analysis of Strawberries

#### 3.5.1. Respiration Rate

The respiration rate of the samples was measured based on a previously reported method [56]. Strawberries were placed in a dryer for 0.5 h with sodium hydroxide (NaOH) solution (0.4 M, 10.0 mL) placed at its base. Then the saturated BaCl_2_ solution (5 mL) and 1% phenolphthalein were added to the solution. The oxalic acid solution (0.2 M) was used for titration to determine the amount of CO_2_ absorbed by the NaOH solution from the strawberries. The same method was used for a control titration. The respiration rate was calculated as follows:(2)$$\mathrm{Respiration rate}=(V_{1}-V_{2})\times M\times 44W\times h\times 100\%$$
where V_1_ is the titration volume of the control group (mL), V_2_ is the titration volume of the sample (mL), M is the oxalic acid concentration (M), 44 represents the molecular weight of CO_2_, W is the sample weight (kg) and h is the determination time. The respiration rate was expressed as mg CO_2_/(kg·h).

#### 3.5.2. Firmness

The firmness of the samples was tested using an FHM-5 texture analyzer (Takemura, Kudamatsu, Japan). Five strawberries from each treatment group were randomly selected then their firmness was measured on the equatorial zone on three sides of each fruit. Puncture tests involved the use of a 6 mm cylinder probe to a penetration depth of 5 mm. 

#### 3.5.3. Weight Loss Percentage

The weight loss percentage (WLP) of the samples was determined by weighing the strawberries daily using an electronic scale (Huazhi, Fuzhou, China). The WLP of strawberries was calculated as the ratio (%) of the weight difference to the initial weight.

#### 3.5.4. Decay Rate

The decay rate of the samples was counted from the number of rotten strawberries each day. The decay rate was calculated as the ratio (%) between the number of rotten strawberries and the initial number.

#### 3.5.5. Color Indices

The change in color value of the strawberries was tested by *L*, *a* and *b* values using a CR-400 Minolta Color Reader (Minolta, Osaka, Japan). This was calibrated using a white plate before use (*L* = 86.3, *a* = 0.3165 and *b* = 0.3142). For these measurements, five strawberries were randomly selected from each group, and each sample was measured three times. The color difference of ΔE was calculated as follow:(3)$$\Delta E=\sqrt{\Delta L^{2}+\Delta a^{2}+\Delta b^{2}}$$
where *L* is the luminance value of the sample, *a* is the redness/greenness value of the sample and *b* is the yellowness/blueness value of the sample.

### 3.6. Biochemical Analysis of Strawberries

#### 3.6.1. Soluble Solids Content

The samples were cut into pieces and homogenized immediately with a blender. After filtration, the juice was collected in sterile conical flasks. The soluble solids content (SSC) values were determined using a PAL-BX/ACID 5 digital refractometer (Atago, Kyoto, Japan). 

#### 3.6.2. Titratable Acidity Content

The titratable acidity (TA) content was determined according to the principle of acid-base neutralization. A total of 10 g of homogenized strawberries were added to 100 mL of deionized water, rested for 30 min and then filtered. Phenolphthalein indicator (1%) were added to 20 mL filtrate, which was then titrated with calibrated NaOH solution (0.1 M). The TA content of samples was calculated according to the consumption of NaOH, and expressed as the mass fraction (%).

#### 3.6.3. Ascorbic Acid Content

The ascorbic acid (AsA) content was determined using the 2,6-dichlorophenol indophenol (DCIP) titration method with modification [57]. A total of 10 g of homogenized strawberries were added to 50 mL of acetic acid solution (1%). The mixture was centrifuged (5000× *g*) at 4 °C for 15 min. Then the supernatant was titrated against the DCIP solution until the reaction liquid turned a pink color which persisted for 30 s. The DCIP solution was standardized using AsA (0.1 g/L). The acetic acid solution (1%) was set as the blank control. The results were expressed as g/100 kg.

#### 3.6.4. Reduced Glutathione Content

Trichloroacetic acid solution (50 g/L) containing EDTA disodium (5M) was added to the samples, which were centrifuged (12,000× *g*) at 4 °C for 20 min. A total of 1.0 mL of supernatant and 1.0 mL of phosphate buffer (0.1 M, pH 7.7) were added to different tubes. DTNB solution (0.5 mL, 4 mM) and phosphate buffer (0.5 mL, 0.1 M, pH 6.8) were added to the corresponding tubes, respectively. The reaction was kept at 25 °C for 10 min. The solution containing distilled water, phosphate buffer and DTNB solution was used as blank control. The absorbance of 412 nm was determined using a spectrophotometer (Shimadzu, Kyoto, Japan). The GSH content was expressed as mmol/kg.

#### 3.6.5. Catalase, Ascorbate Peroxidase and Superoxide Dismutase Activity

Extraction buffer was added to the samples, which were ground, homogenized and then centrifuged (12,000× *g*) at 4 °C for 30 min. The supernatant was the enzyme extract. Distilled water was set the control. The CAT, APX and SOD activities were determined according to the instructions of the enzyme activity kit. The activity unit for CAT (0.01 ΔOD_240_·min^−1^·g^−1^) and APX (0.01 ΔOD_290_ ·min^−1^·g^−1^) were defined as a 0.01 decrease in the absorbance value of the reaction system per gram of the sample at wavelength of 240 nm and 290 nm, respectively. One SOD activity unit (U) was defined as a 50% inhibition of NBT photochemical by the reaction system per gram strawberry per min.

### 3.7. Statistical Analysis

The one-factor analysis of variance (ANOVA) and Duncan’s test were used for multiple comparisons by SPSS 22 (IBM, New York, NY, USA). The difference was considered to be statistically significant if *p* < 0.05. The figures were drawn with R package of ggplot2 developed by Wickham [58]. 

## 4. Conclusions

In this study, GA was successfully grafted into CMCS catalyzed by recombinant bacterial laccase from *Streptomyces coelicolor* as determined by measuring amino acid content, UV-Vis and FTIR spectroscopy. As the research has demonstrated that the GA-g-CMCS coating helped protect the antioxidant properties and nutrients of strawberries and prolonged shelf life relative to the control group. The 1.5% GA-g-CMCS coating reduced the WLP of strawberries during storage (initially 12.7%) to 8.4% and the decay rate from 36.7% to 8.9%. It also reduced the nutrient loss, including SSC, TA and ASA, and maintained the antioxidant enzyme activity, such as keeping TA content above 60%. Future studies are needed to optimize the optimization of catalytic conditions for the synthesis of CMCS conjugated with phenolic acids for industrial applications. Bacterial laccase-catalyzed CMCS grafted phenolic acid is simpler to operate than chemical methods, can be used under milder conditions, and is safe and effective, which is conducive to the application of fruit preservation and provides a reference method for the synthesis and application of subsequent CMCS derivatives. 

## Figures and Tables

**Figure 1 molecules-27-08496-f001:**
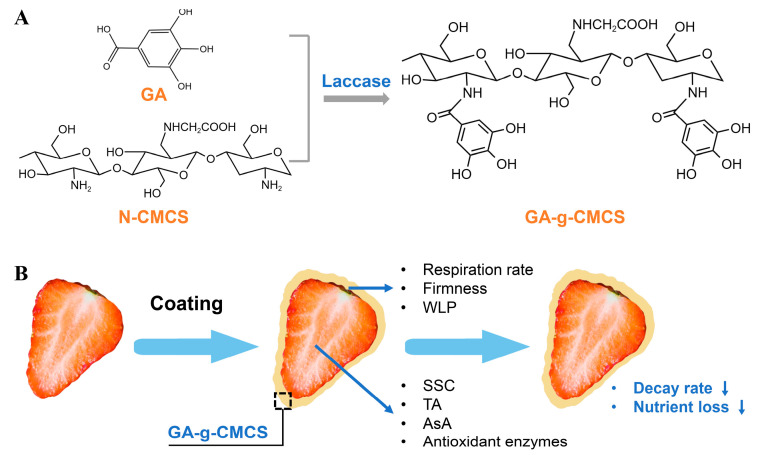
Mechanism of CMCS and GA grafting and working for strawberry preservation. (**A**) The mechanism of synthesis of GA-g-CMCS by laccase-catalyzed grafting reaction. (**B**) The schematic diagram of strawberry preservation treated with GA-g-CMCS coating. GA: gallic acid, N-CMCS: N-carboxymethyl chitosan, GA-g-CMCS: GA grafted onto N-CMCS, WLP: weight loss percentage, SSC: Soluble solids content, TA: titratable acidity, AsA: ascorbic acid.

**Figure 2 molecules-27-08496-f002:**
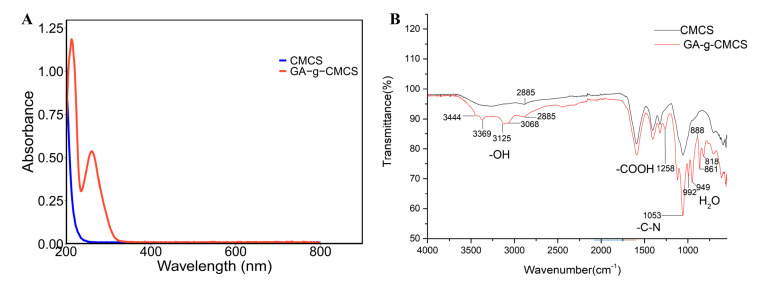
(**A**) UV-vis and (**B**) FTIR spectrogram of CMCS and GA-g-CMCS.

**Figure 3 molecules-27-08496-f003:**
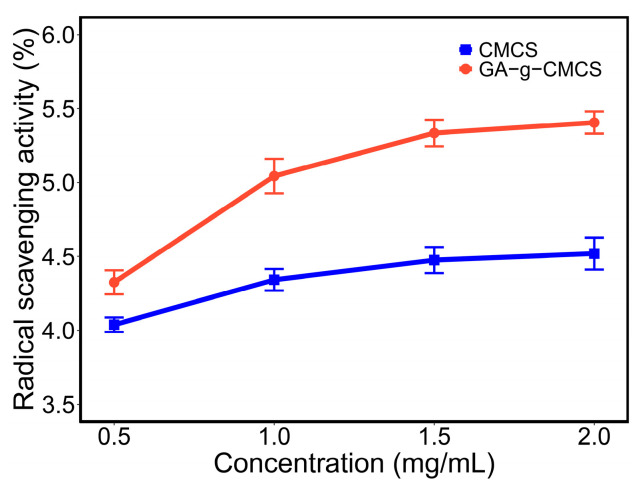
DPPH free radical scavenging capacity of CMCS and GA-g-CMCS.

**Figure 4 molecules-27-08496-f004:**
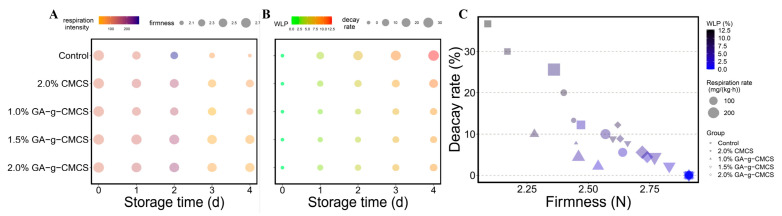
(**A**) Variation in respiration rate and firmness, (**B**) WLP and decay rate and (**C**) decay rate relative to firmness of strawberries at storage time of 0, 1, 2, 3 and 4 days.

**Figure 5 molecules-27-08496-f005:**
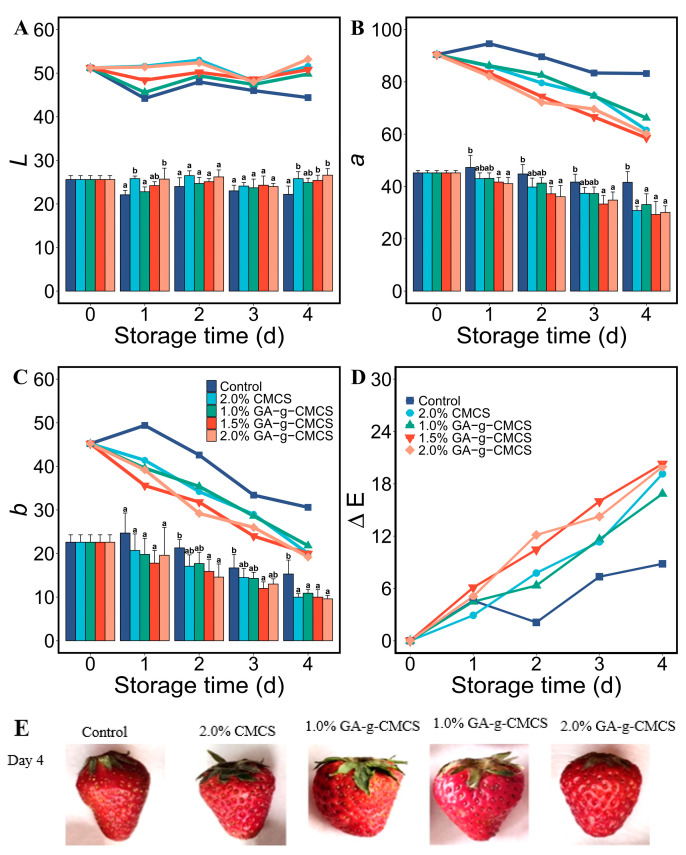
(**A**) Variation in luminance *L*; (**B**) color value of red–green axis *a*; (**C**) color value of blue-yellow axis *b*; and (**D**) color difference ΔE at storage time of 0, 1, 2, 3 and 4 days. (**E**) Photograph of strawberry coated with pure deionized water, CMCS and GA-g-CMCS, respectively.

**Figure 6 molecules-27-08496-f006:**
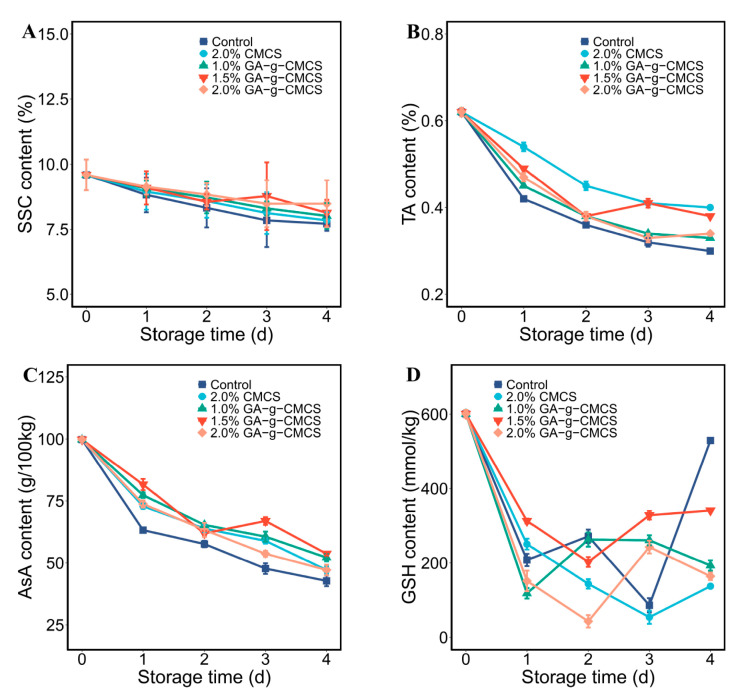
(**A**) Variation in SSC; (**B**) TA; (**C**) AsA; and (**D**) GSH (of strawberries at storage time of 0, 1, 2, 3 and 4 days).

**Figure 7 molecules-27-08496-f007:**
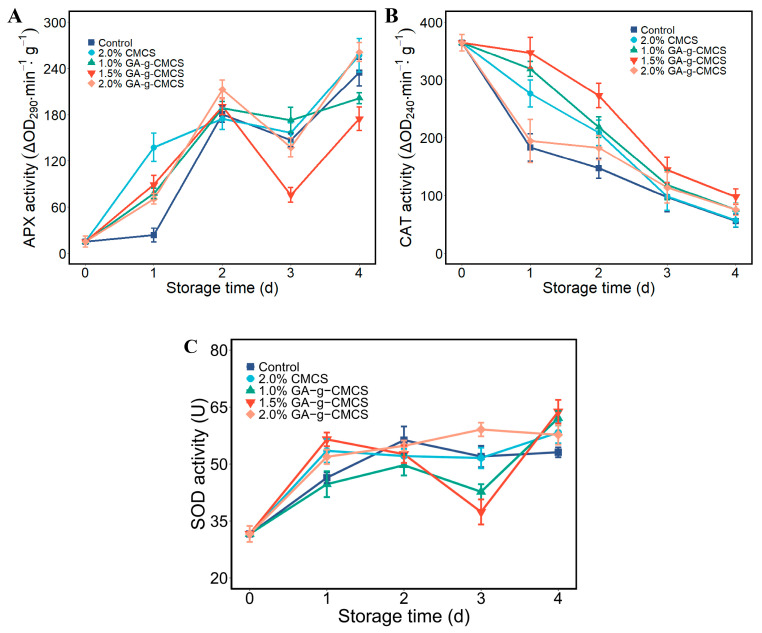
(**A**) Variation in CAT activity; (**B**) APX activity and (**C**) SOD activity of strawberries at storage time of 0, 1, 2, 3 and 4 days.

## Data Availability

Not applicable.

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
