# Peer review of "Enhancement of Antioxidant Property of N-Carboxymethyl Chitosan and Its Application in Strawberry Preservation"

_molecules, 2022, doi:10.3390/molecules27238496_

Round 1
Reviewer 1 Report
The manuscript by Luyao Bian et al. in an interesting investigation about the strawberry preservation activity of a N-carboxymethyl chitosan-grafting gallic conjugate obtained by Laccase catalysis. The topic of the manuscript is worthy of investigation and fits with the scope of the journal. It is an opinion of this reviewer that the paper can be eventually published, but before further processing some key issues should be addressed as suggested below.
Abstract
Some more key results should be added here. In its current form the section is too literal
Introduction
The presentation of novelty of the study should be improved. Authors should clarify what’s new in their approach (e.g. the synthetic strategy, the application) and what is the expected advantages of their proposed strategy
Results and discussion
Authors should elucidate why only DPPH test was used to assess the antioxidant capacity
Authors should compare their results with some available literature data to strengthen the novelty and the advantages of their approach
Figure 1. can authors explain why the formation of an amide bond between CMCS and GA is indicated as the result of the laccase catalysis
Materials and methods
Authors should indicate how was the synthesis of the conjugation optimized
References
Please check the references list. There is no correspondence between text and references.
Reviewer 2 Report
This work is very complete and suitable for Molecules. The authors perform the following determinations: Determination of amino content, UV-Vis and FTIR analyses, Antioxidant activity, Respiration rate, Firmness, Weight loss, Decay rate, Color indices, Titratable acidity content, Ascorbic acid content, . Reduced glutathione content, Catalase, and ascorbate peroxidase and superoxide dismutase activity. These determinations are very adequate and sufficient for this paper. It would be necessary for the future to carry out a thermal analysis of GA-g-CMCS, another important fact is to evaluate its toxicity, its antimicrobial capacity and its biodegradability.
Author Response
Thanks very much for your positive evaluation of our work.
Reviewer 3 Report
1- page 1 line 1: the title shoud be summrized to Enhancement of antioxidant property of N-carboxymethyl chi- 2 tosan and its application in strawberry preservation instead of Enhancement of antioxidant property of N-carboxymethyl chi- 2 tosan through grafting gallic acid catalyzed by small laccase 3 and its application in strawberry preservation
2-page 1 line 17 what the is the value of weight loss ?
3- page 2 line 28,31 what is the meaning of ROS?
4- page 4 line 112 what is the meaning of DPPH
5- page 11 line 374 the meaning of abriviations of L,a,b in equation 3 should be written under the equations
6- conclusion should be improved
Round 2
Reviewer 1 Report
Authors well addressed all the commnents and modified the manuscript accordingly. The paper can now be published
Author Response

(The authors gave the same response as above.)
